# Dietary γ-Aminobutyric Acid Supplementation Inhibits High-Fat Diet-Induced Hepatic Steatosis via Modulating Gut Microbiota in Broilers

**DOI:** 10.3390/microorganisms10071281

**Published:** 2022-06-24

**Authors:** Qu Chen, Dan Hu, Xiaoting Wu, Yuyan Feng, Yingdong Ni

**Affiliations:** 1Key Laboratory of Animal Physiology & Biochemistry, Nanjing Agricultural University, Nanjing 210095, China; 2018207007@njau.edu.cn (Q.C.); 2020107017@njau.edu.cn (D.H.); 2020807160@njau.edu.cn (X.W.); 2021207058@njau.edu.cn (Y.F.); 2MOE Joint International Research Laboratory of Animal Health & Food Safety, Nanjing 210095, China

**Keywords:** broilers, γ-aminobutyric acid, lipid metabolism, gut microbiota

## Abstract

The present study aims to investigate the effect of γ-aminobutyric acid (GABA) on liver lipid metabolism and on AA broilers. Broilers were divided into three groups and fed with low-fat diets, high-fat diets, and high-fat diets supplemented with GABA. Results showed that GABA supplementation decreased the level of triglyceride (TG) in the serum and liver of broilers fed high-fat diets, accompanied by up-regulated mRNA expression of genes related to lipolysis and β-oxidation in the liver (*p* < 0.05). Furthermore, GABA supplementation increased liver antioxidant capacity, accompanied by up-regulated mRNA expression of antioxidant genes (*p* < 0.05). 16S rRNA gene sequencing showed that GABA improved high-fat diet-induced dysbiosis of gut microbiota, increased the relative abundance of Bacteroidetes phylum and *Barnesiella* genus, and decreased the relative abundance of Firmicutes phylum and *Ruminococcus_torques_group* and *Romboutsia* genus (*p* < 0.05). Moreover, GABA supplementation promoted the production of propionic acid and butyric acid in cecal contents. Correlation analysis further suggested the ratio of Firmicutes/Bacteroidetes negatively correlated with hepatic TG content, and positively correlated with cecal short chain fatty acids content (*r* > 0.6, *p* < 0.01). Together, these data suggest that GABA supplementation can inhibit hepatic TG deposition and steatosis via regulating gut microbiota in broilers.

## 1. Introduction

Lipids (oil and fat) are necessary for poultry growth, not only providing essential fatty acids but the main source of feed energy. Therefore, lipids were usually added to poultry diets to improve the feed reward and the growth performance [1,2]. However, there are some factors restricting the improvement of dietary lipid levels in poultry diets. For example, incomplete absorption of dietary lipids by poultry will lead to excessive deposition of body fat and reduce carcass quality; moreover, a high-fat diet may also increase the burden of liver lipid metabolism, leading to liver injury and metabolic disorders [3].

In the intensive feeding system, broilers are often fed with high-energy diets to satisfy their rapid growth. Unlike mammals, in chickens, fatty acids absorbed by the intestinal tract are directly transported to the liver through the portal vein [4], and more than 90% of body fat is synthesized in the liver [5,6]. Excessive fat deposition in the chicken liver will cause hepatic steatosis, food intake depression, and lower growth performance, and will even lead to death [7].

γ-aminobutyric acid (GABA) is a non-protein amino acid that acts as an inhibitory neurotransmitter in the central nervous system of animals, moreover, it’s generally recognized as a safe and green feed additive [8]. Many studies showed that GABA has many biological functions, including antioxidant, antidiabetic, and immune-modulating properties [9,10,11]. Oxidative stress and gut microbiota disorders are considered to contribute to the development of hepatic steatosis and metabolic abnormalities in poultry [12,13]. Recent studies on rodent models suggest that GABA can prevent high-fat diet-induced obesity by improving oxidative stress and gut microbial structure [14,15]. This evidence indicated that GABA can regulate lipid metabolism and reduce liver lipid deposition. Given that the digestive system and gut microbiota composition of chickens are different from that of rodents, the effect and regulatory mechanism of GABA on lipid metabolism in broilers need further investigation. Therefore, this study is aimed to explore the effect of GABA on liver lipid deposition, oxidative stress, cecal short-chain fatty acids (SCFAs), and gut microbiota in broiler chickens, which will provide a good understanding of GABA in alleviating hepatic steatosis in broiler chickens.

## 2. Materials and Methods

### 2.1. Animals, Diets, and Experimental Protocol

All animal procedures were permitted by the Institutional Animal Care and Use Committee of Nanjing Agricultural University according to the Guidelines on Ethical Treatment of Experimental Animals (2006) No. 398 set by the Ministry of Science and Technology (2006, Beijing, China).

A total of 120 male 1-day-old AA broilers were randomly assigned to 3 groups, with 8 cages in each group and 5 birds in each cage. Broilers were fed one of the following dietary treatments: a low-fat diet (Con, 4.5% calculated total fat) and a high-fat diet (HF, 9.0% calculated total fat) supplemented with or without 100 mg/kg of GABA (FG, 9.0% calculated total fat + 100 mg/kg GABA) (Degao Biotechnology Co., Ltd., Jinan, China), the added fat was soybean oil. Broilers were given free access to both food and water throughout the experimental time. Diets were formulated to meet or exceed NRC (1994) requirements. The diet composition is presented in Table 1. All broilers were housed in an environmentally controlled house. The temperature was maintained at 34 °C during the first week and then reduced by 3 °C/week until it reached 25 °C.

On day 40, 8 broilers from each treatment were randomly selected from different cages and slaughtered after a 12 h fast. Blood samples were collected using 5 mL tubes without anticoagulants. The liver tissues at the same site were collected and soaked in 4% paraformaldehyde for H&E and oil red O staining. The other liver tissues were frozen immediately in liquid nitrogen and stored at −80 °C for further analysis. The cecal contents were collected aseptically, snap-frozen, and stored at −80 °C for 16S rRNA sequencing and SCFAs analysis.

### 2.2. Staining for Histopathology

The liver tissues were cut and fixed with 4% paraformaldehyde and were embedded in paraffin. Sections were stained with hematoxylin−eosin (H&E) or oil red O to investigate architecture. Stained slides were scanned with the Pannoramic SCAN II, and images were captured with 3DHISTECH software (3DHISTECH, Ltd., Budapest, Hungary).

### 2.3. Measurement of Biochemical Parameters in Serum and Liver

Serum biochemical parameters, including TG, TC, LDL-C, and HDL-C, were measured using an automatic biochemical analyzer (7020, HITACHI, Tokyo, Japan). Hepatic TG content was determined using a commercial kit provided by Applygen Co. Ltd. (Beijing, China), strictly following the manufacturer’s instructions.

### 2.4. Biochemical Analysis of Oxidative and Antioxidative Biomarkers in Liver

Liver tissues were homogenized in 0.9% NaCl buffer and centrifuged at 2500 rpm for 10 min at 4 °C. The supernatants were collected for the determination of oxidative products and antioxidant capacity. Protein concentration was determined using a BCA protein assay kit. Hepatic catalase (CAT, cat. no. A009), malonaldehyde (MDA, cat. no. A003-1), and glutathione peroxidase (GSH-Px, cat. no. A005) were determined using commercial kits (Jiancheng Co. Ltd., Nanjing, China), strictly following the manufacturer’s instructions.

### 2.5. RNA Isolation, cDNA Synthesis, and Real-Time PCR

Total RNA was extracted from liver samples with TRIzol reagent (Tsingke, China, TSP401). The concentration and quality of the RNA were measured with a NanoDropND-1000 Spectrophotometer (Thermo Fisher Scientific, Madison, WI, USA). Then 1 μg of total RNA was treated with ABScript III RT Master Mix (ABclonal, Wuhan, China, RK20429) according to the manufacturer’s instructions. Quantitative Real-time PCR was performed using 1 μL first-strand cDNA with the Genious 2X^®^ SYBR Green Fast qPCR Mix (ABclonal, China, RK21206) in a final volume of 10 μL. All samples were run in triplicate and underwent 45 amplification cycles in an Mx3000P (Stratagene, Santa Clara, CA, USA). Peptidylprolyl isomerase A (PPIA), which is not affected by the experimental factors, was chosen as the reference gene. All the primers listed in Table 2 were synthesized by Tsingke Company (Nanjing, China). The method of 2^−ΔCT^ t was used to analyze the real-time PCR results, and gene mRNA levels were expressed as the fold change compared to the mean value of the control group.

### 2.6. Cecal SCFAs Analysis

Cecal chyme samples were mixed evenly with double steam water at 1:5 and centrifuged at 2000 rpm for 12 min at 4 °C. The supernatants (1 mL) were collected and added to 0.2 mL of deproteinizing-acidifying solution (metaphosphoric acid [25% *w*/*v*] and crotonic acid [0.65% *w*/*v*]) for SCFAs measurement using a gas chromatography method (Agilent 7890B, Agilent, CA, USA). The samples were separated on a fused silica capillary column (Agilent, Santa Clara, CA, USA) at 110 °C, 3 min; 110–150 (40 °C/min) column temperature program, a 200 °C injection temperature, and 220 °C FID detector temperature with N2 as a carrier gas.

### 2.7. Analysis of 16S rRNA Sequencing’ in Cecal Microbiome

Total genome DNA from samples was extracted using CTAB/SDS method. DNA concentration and purity were monitored on 1% agarose gels. According to the concentration, DNA was diluted to 1 ng/µL using sterile water. 16S rRNA genes of distinct regions V3 and V4 were amplified using specific primers (338F: 5′-ACTCCTACGGGAGGCAGCAG-3′, 806R: 5′-GGACTACHVGGGTWTCTAAT-3′). The library was sequenced on an Illumina MiSeq PE300 platform/NovaSeq PE250 platform (Illumina, San Diego, CA, USA) according to the standard protocols by Majorbio Bio-Pharm Technology Co., Ltd. (Shanghai, China). Paired-end reads were merged using FLASH (V1.2.7, http://ccb.jhu.edu/software/FLASH/, accessed on 20 June 2022). The filtered reads were clustered into operational taxonomic units (OTUs) with a 97% sequence identity with UPARSE (version 7.1, http://www.drive5.com/uparse/, accessed on 20 June 2022). Chimera was removed during clustering. The OTU representative sequence was analyzed with the RDP Classifier (version 2.2, http://rdp.cme.msu.edu/, accessed on 20 June 2022) against the Silva 16S rRNA database at a confidence threshold of 70%. Correlation analyses between the abundances of microbiota, the hepatic TG content, and the cecal SCFAs content were conducted by Spearman’s correlation analysis.

### 2.8. Statistical Analysis

Data regarding growth performance, SCFAs, biochemical parameters, and oxidative and antioxidative biomarkers were analyzed by one-way ANOVA with a Bonferroni post hoc test using SPSS software packages (SPSS version 20.0 for Windows, IBM, Armonk, NY, USA). The data regarding gut microbiome were analyzed by Kruskal-Wallis with a Turkey-Kramer post hoc test. *p*-values < 0.05 were considered to be statistically significant.

## 3. Results

### 3.1. Growth Performance

As presented in Figure 1A, from 1 to 35 d, there was no significant difference in feed intake of broilers in each group (*p* > 0.05). From 1 to 14d broilers fed high-fat diets had a lower feed conversion ratio compared with the broilers fed low-fat diets (*p* < 0.05) (Figure 1B). On 14 and 40 d, broilers fed high-fat diets had heavier body weight compared with the broilers fed low-fat diets; compared with the Con and FG groups, the body weight of the broilers in the HF group increased significantly on 21 d (*p* < 0.05) (Figure 1C).

### 3.2. Serum Lipid Profile

As shown in Table 3, compared with low-fat diets, high-fat diets increased TG levels in the serum on 40 d (*p* < 0.05), whereas GABA supplementation significantly alleviated the increase of serum TG caused by high-fat diets (*p* < 0.05). There was no significant difference in the concentrations of TC, HDL-C, and LDL-C in each group (*p* > 0.05).

### 3.3. Liver Histology, TG Content, and Lipid Metabolism-Related Gene Expression

As shown in Figure 2A, H&E and oil red O staining of the liver sections showed that high-fat diets induced obvious hepatocyte vacuolation and increased liver lipid droplets accumulation, while GABA supplementation significantly reduced hepatic steatosis in broilers fed high-fat diets. Correspondingly, TG content in the liver tissues significantly increased in the HF group compared with the Con and FG groups (*p* < 0.05) (Figure 2B). To investigate whether GABA supplementation regulated hepatic lipid accumulation through modulating the gene expression implicated in fatty acid metabolism, the mRNA expression of genes implicated in fatty acid synthesis (*FAS*, *ACC*, and *SREBP1*), lipid droplet formation (*PLIN1*), fatty acid transport (*FATP1*), lipolysis (*ATGL* and *LPL*) and fatty acid β-oxidation (*PPARα* and *CPT1α*) was determined (Figure 2C). The mRNA level of *PLIN1* in the HF group was significantly increased compared to the Con and FG groups (*p* < 0.01). In contrast, the expression of hepatic ATGL was significantly decreased in the HF group compared to the Con and FG groups (*p* < 0.05). The expression levels of hepatic *LPL*, *PPARα*, and *CPT1α* were significantly increased in the GABA supplementation broilers compared to those of the Con and HF groups (*p* < 0.05). In addition, there was no significant difference in the mRNA levels of *FAS*, *ACC*, and *SREBP1* among the three groups (*p* > 0.05).

### 3.4. Liver Antioxidant Indexes and Antioxidant-Related Gene Expression

High-fat diets significantly promoted the production of hepatic MDA in the HF group compared to the Con group, while GABA supplementation decreased hepatic MDA levels compared to the HF group (*p* < 0.01) (Figure 3A). Moreover, the content and activity of the antioxidant enzymes GSH-Px (*p* = 0.059) and CAT (*p* < 0.01) decreased in the HF group compared to the Con group, while GABA supplementation increased the levels of GSH-Px and CAT compared to the HF group (*p* < 0.05) (Figure 3B,C). The mRNA expression of genes implicated in antioxidants showed that the level of *NRF2* decreased in the HF group compared to the Con (*p* = 0.06) and FG groups (*p* < 0.01) (Figure 3D). GABA supplementation significantly increased the mRNA levels of GPX1, HO1, and SOD in the liver compared to the Con and HF groups (*p* < 0.05) (Figure 3D).

### 3.5. Community Diversity and Composition of Cecal Microbiota

A principal coordinate analysis (PCoA) was performed to assess similarities and differences among groups. Samples collected from the HF group were significantly separated from those of the Con group in the PCoA analysis on the OTU, order, and genus levels, on the phylum level, they were mildly separated. The microbial community of chickens in the FG group was between the Con and FG groups at the OTU, phylum, order, and genus levels (Figure 4A–D). There was no significant difference in the ACE and Shannon indices among groups (Figure 4E,F). However, the Simpson index was significantly reduced in the HF group compared to the Con group (*p* < 0.05) (Figure 4G).

Taxonomic profiling indicated that the relative abundance of the phylum Firmicutes and Bacteroidetes was altered obviously in the HF group. Firmicutes significantly increased while Bacteroidetes decreased in the HF group compared to the Con and FG groups (*p* < 0.05) (Figure 5A). At the order level, the relative abundance of Oscillospirales and Lanchnospirales increased while Bacteroidales decreased in the HF group compared to the Con group, and supplementation with GABA reversed the reduction in the abundance of Bacteroidales in broilers fed high-fat diets (*p* < 0.05) (Figure 5B). At the genus level, the beneficial bacteria *Barnesiella* were significantly reduced while the abundance of *Ruminococcus_torques_group* and *Romboutsia* increased in the HF group compared to the Con and FG groups (*p* < 0.05) (Figure 5C).

Linear discriminant analysis effect size (LEfSe, LDA score > 4) analysis showed that the genus *Barnesiella*, family Barnesiellaceae, order Bacteroidales, class Bacteroidia, and phylum Bacteroidetes were enriched in the Con group (Figure 5D). The genus *Ruminococcus_torques_group* and *Romboutsia*, family Lactobacillaceae, Oscillospiraceae and Peptostreptococcaceae, order Lachnospirales, Oscillospirales and Peptostreptococcales-Tissierellales, class Clostridia, phylum Firmicutes predominated in the HF group (Figure 5D).

### 3.6. Cecal SCFAs Concentrations

To further examine whether GABA affected the cecal bacterial metabolites, we measured cecal SCFAs in broiler chickens. As shown in Figure 6, GABA supplementation significantly increased the levels of propionic acid, butyric acid, and total SCFAs compared with the HF group (*p* < 0.05) (Figure 6B,C,G). The concentrations of butyric acid and isovaleric acid in the FG group were higher than those in the Con group (*p* < 0.05) (Figure 6C,E).

### 3.7. Correlation Analysis

In order to investigate whether the altered gut microbiota affected the hepatic lipid accumulation and cecal SCFAs concentrations, we performed correlation analyses between the relative abundances of cecal microbiota and the levels of hepatic TG and cecal SCFAs. As presented in Figure 7, the Bacteroidetes and *Barnesiella* abundance were negatively correlated with the hepatic TG content (*p* < 0.01). By contrast, the Firmicutes, *Ruminococcus_torques_group*, *Romboutsia* abundance, and the ratio of Firmicutes/Bacteroidetes were positively correlated with the hepatic TG content (*p* < 0.05). In addition, the ratio of Firmicutes/Bacteroidetes was negatively correlated with the cecal total SCFAs concentrations, while *Barnesiella* abundance was positively correlated with the cecal total SCFAs and butyric acid concentrations (*p* < 0.01).

## 4. Discussion

GABA is a metabolite of glutamic acid and is widely used as a nutritional agent in the field of poultry production to improve the resistance to stress. Under different conditions, the effects of GABA on the growth performance of broilers were not consistent. GABA supplementation can significantly decrease the heterophil-lymphocyte ratio (H/L ratio) and increase body weight gain in broilers suffering from chronic heat stress [16]. However, dietary GABA reduced stress indices including corticosterone and H/L ratio without reversing stocking density-induced growth depression [17]. A study in a mouse model showed that GABA treatment not only prevented weight gains in high-fat diets fed mice, but also markedly decreased plasma TG without affecting feed intake [15]. Our results showed that GABA supplementation reduced serum TG levels without affecting the growth-promoting effect of high-fat diets on broilers.

Lipid accumulation in the form of TG is considered to be a representative index of a tissue’s exposure to fatty acids [18]. Our data showed that GABA supplementation significantly decreased serum and hepatic TG contents. The reduced hepatic lipid deposition was also evidenced by histological staining, the H&E and oil red O sections of liver tissues further confirmed that GABA can alleviate hepatic steatosis caused by high-fat diets. To elucidate the mechanism of the lipid-lowering effects of GABA, we analyzed the mRNA expression of genes related to lipid metabolism in the liver of broiler chickens. ATGL is the rate-limiting enzyme catalyzing the first step of TG breakdown to fatty acids and glycerol [19]. LPL is a rate-limiting enzyme that catalyzes the hydrolysis of serum TG into fatty acids, which are subsequently taken into tissues [20]. A decreased ATGL and LPL expression could increase serum TG and cause excessive fat deposition in the liver [19,21]. PPARα and CPT1α are key mediators in the control of fatty acid β-oxidation [22]. Our data showed that compared with the HF group, increased expressions of ATGL, LPL, PPARα, and CPT1α were found in broilers treated with GABA, moreover, there was no significant difference in the expression of genes related to de novo fatty acid synthesis among groups. Overall, the current studies demonstrated that high-fat diets promoted exogenous lipid deposition in the liver, and GABA supplementation could regulate lipid metabolism in the liver of broilers partially by promoting lipolysis and fatty acid β-oxidation.

The pathological mechanisms of fatty liver in chickens are similar to nonalcoholic fatty liver disease (NAFLD), including insulin resistance, inflammation, and liver oxidative stress [23]. The multiple-hit pathogenesis theory considered that long-term hepatic fat accumulation can produce lipotoxicity, resulting in impairment of hepatocyte function and reduction of antioxidant capacity [24]. The improvement of the antioxidant capacity of hepatocytes can effectively reduce liver lipid deposition [25]. Previous studies suggested that GABA supplementation can reduce hepatic MDA content and induce higher GSH-Px activity in broiler chickens suffering from heat stress [16]. Our data also showed that GABA supplementation reduced the content of MDA (a product of lipid peroxidation of membrane), and improved the activity, content, and gene expression of liver antioxidant enzymes in broilers fed high-fat diets. Moreover, the mRNA expression of nuclear factor E2-related factor 2 (*NRF2*), a central regulator for the expression of antioxidant and detoxifying enzymes [26], was increased in broilers supplemented with GABA. Our results indicated that GABA induced the expression of NRF2 to stimulate transcription of downstream antioxidant enzymes, thus contributing to the amelioration of high-fat diet-induced liver oxidative stress and steatosis.

In recent years, many studies demonstrated that the altered gut microbiota induced by dietary treatments exert important roles in regulating lipid metabolism [27,28]. Our PCoA analysis showed that the microbial community of broilers fed high-fat diets was different from that of broilers fed low-fat diets. Simpson index to assess diversity was also significantly different between the HF and Con groups, suggesting a deep alteration in the gut microbiome. Compared with the HF group, the microbial community of the FG group was closer to that of the Con group, suggesting that GABA supplementation improved intestinal microflora dysbiosis. At the phylum level, GABA supplementation increased the relative abundance of Bacteroidetes and reduced the ratio of Firmicutes/Bacteroidetes in broilers fed high-fat diets. Epidemiologic data suggested that the gut Firmicutes/Bacteroidetes ratio highly positively correlated with hepatic steatosis [29]. At the genus level, the FG group showed a higher abundance of *Barnesiella* and a lower abundance of *Ruminococcus_torques_group* and *Romboutsia* than the HF group. *Barnesiella* has an immunomodulatory effect, which can prevent the colonization of drug-resistant pathogens in the intestine [30]. The increase of *Barnesiella* abundance is beneficial to improving the resistance of broilers to lipopolysaccharide challenge, alleviating inflammatory response [31]. Inflammation is an important factor inducing and aggravating hepatic steatosis in chicken [32]. In our study, *Barnesiella* abundance is negatively correlated with hepatic TG content, suggesting that it may alleviate liver lipid deposition by regulating immune function. Similar to the previous study on rats, increased *Romboutsia* abundance induced by high-fat diets promoted liver lipid deposition and steatosis [33]. A recent study found that *Ruminococcus_torques_group* may be a key intestinal bacterium in promoting lipid deposition in poultry. Lyu et al., speculated that the decrease of *Ruminococcus_torques_group* abundance was helpful to improve animal lipid metabolism and reduce fat deposition [34].

Bacteroidetes exerts an important role in carbohydrate fermentation and can ferment carbohydrate to produce a variety of SCFAs [35]. *Barnesiella*, which is wildly considered a probiotic, is also a SCFAs-producing bacteria [36]. Consistently, GABA supplementation significantly increased cecal total SCFAs, propionic acid, and butyric acid concentrations. Si et al., found that supplemented obese rats with GABA-enriched rice bran increased propionate and butyrate production by promoting associated synthesizing enzymes in gut microbiota [14]. Li et al., found that feed supplementation with propionate can inhibit fat deposition in broilers by reducing feed and caloric intake, but not via direct regulation of hepatic fatty acid synthesis; moreover, propionate may improve the composition of intestinal microbiota [27]. Butyrate not only acts as an energy source for intestinal epithelial cells but also has a variety of biological functions. Sun et al., found that supplemented high-fat diets fed rats with butyrate significantly reduced TG deposition and oxidative stress. It can be attributed to the fact that butyrate acted as a histone deacetylase inhibitor to up-regulate PPARα and NRF2 expression with enhanced H3K9Ac modification on their promoters, which ultimately improved liver β-oxidation and antioxidation capacity [25,37]. These studies were highly consistent with our data and suggested that GABA may act as a prebiotic to maintain microbiome homeostasis and increase SCFA production, which contributes to improving liver lipid metabolism in broilers.

## 5. Conclusions

In conclusion, dietary GABA supplementation increased liver antioxidant capacity and decreased liver steatosis and TG deposition in the broilers fed high-fat diets, and these effects might be associated with alterations in the composition of gut microbiota. These findings suggest that GABA can be used as a feed additive to alleviate the dysbiosis of gut microbiota and suppress liver steatosis in broiler chickens fed high-fat diets.

## Figures and Tables

**Figure 1 microorganisms-10-01281-f001:**
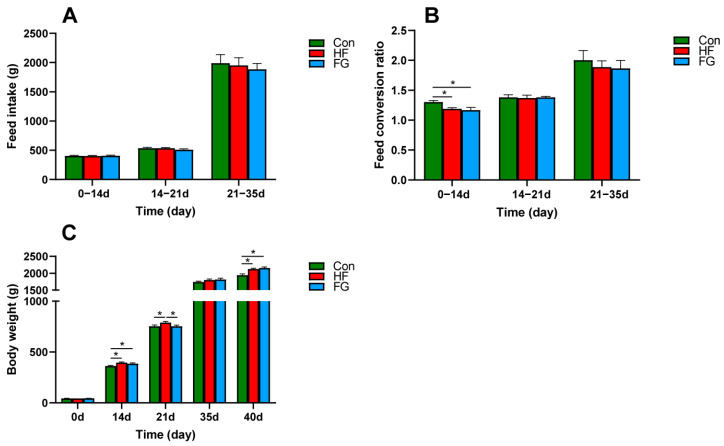
Effects of dietary fat level and GABA supplementation on feed intake (**A**), feed conversion ratio (**B**), and body weight (**C**). Data are presented as mean ± SEM, * *p* < 0.05.

**Figure 2 microorganisms-10-01281-f002:**
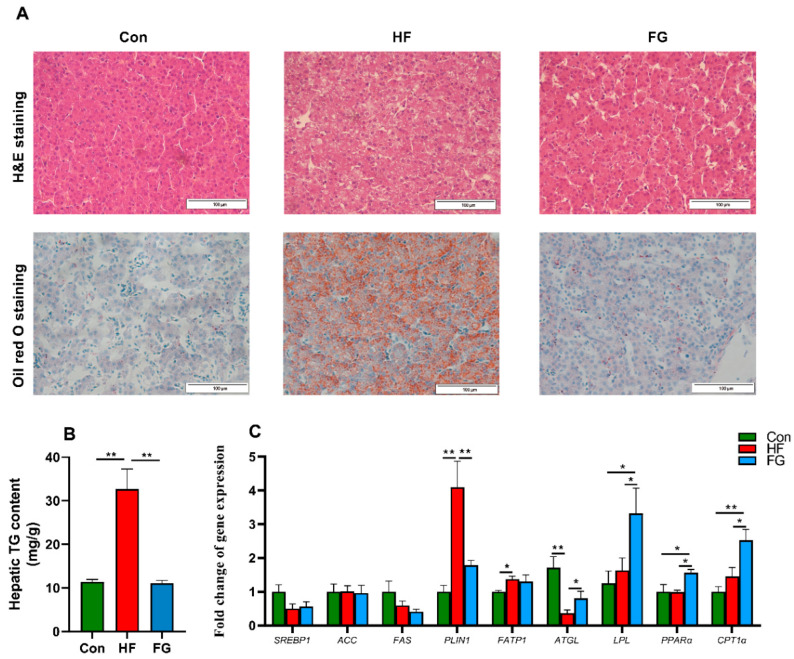
Effects of dietary fat level and GABA supplementation on histological changes (**A**), hepatic TG content (**B**), and the expression of lipid metabolism-related genes (**C**). Data are presented as mean ± SEM, * *p* < 0.05, ** *p* < 0.01. TG, triglyceride; *SREBP1*, sterol regulatory element-binding protein 1; *ACC*, acetyl-CoA carboxylase; *FAS*, fatty acid synthase; *PLIN1*, Perilipin 1; *FATP1*, fatty acid transport protein 1; *ATGL*, adipose triglyceride lipase; *LPL*, lipoprotein lipase; *PPARα*, peroxisome proliferator-activated receptor α; *CPT1α*, carnitine palmitoyl transferase 1α.

**Figure 3 microorganisms-10-01281-f003:**
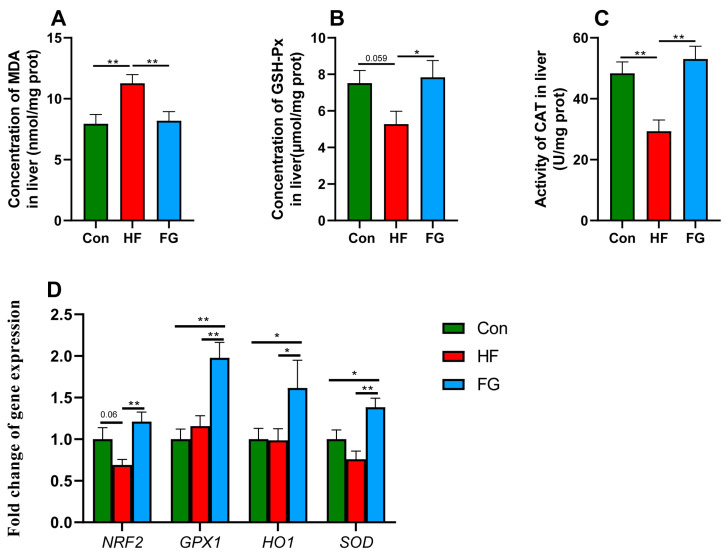
Effects of dietary fat level and GABA supplementation on liver MDA content (**A**), GSH-Px content (**B**), CAT activity (**C**), and the expression of antioxidant-related genes (**D**). Data are presented as mean ± SEM, * *p* < 0.05, ** *p* < 0.01. MDA, malondialdehyde; GSH-Px, glutathione peroxidase; CAT, catalase; *NRF2*, nuclear factor E2-related factor 2; *GPX1*, glutathione peroxidase 1; *HO1*, heme oxygenase 1; *SOD*, superoxide dismutase.

**Figure 4 microorganisms-10-01281-f004:**
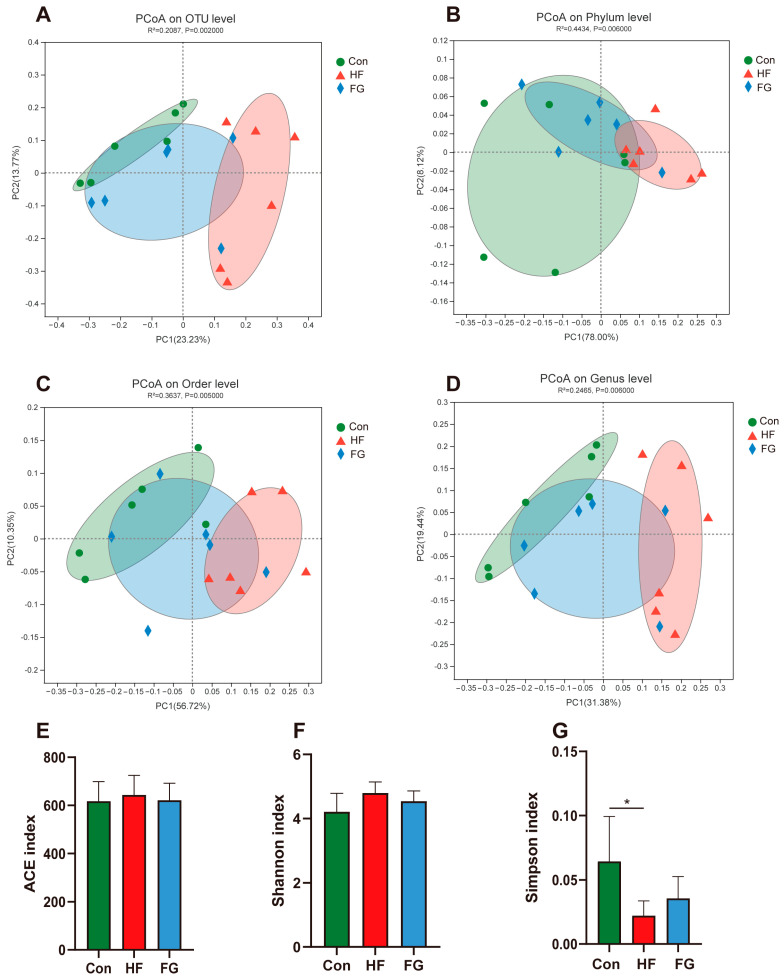
Analysis of intestinal microbial community diversity. Principal co-ordinates analysis (PCoA) on OTU level (**A**), phylun level (**B**), order level (**C**), genus level (**D**). Histogram for comparison of OTU diversity, ACE index (**E**), Shannon index (**F**) Simpson index (**G**). Data are presented as mean ± SD, * *p* < 0.05.

**Figure 5 microorganisms-10-01281-f005:**
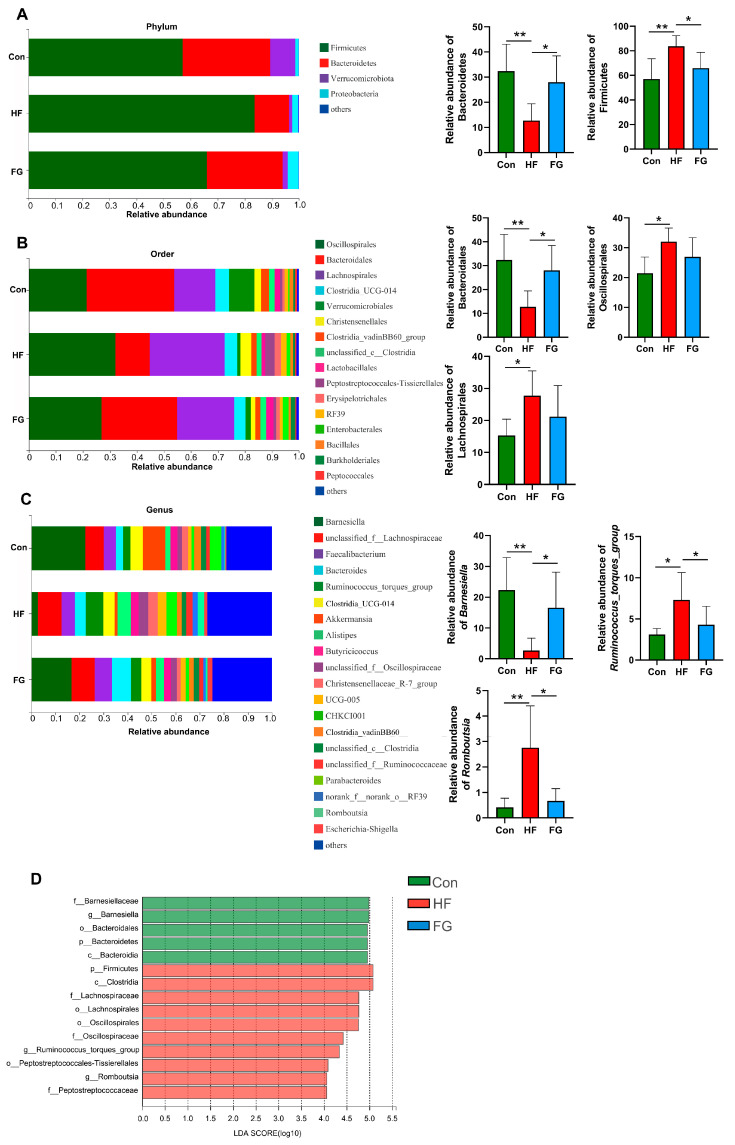
Effects of dietary fat level and GABA supplementation on the cecal microbiota. Microbiota compositions at the phylum level (**A**), microbiota compositions at the order level (**B**), microbiota compositions at the genus level (**C**). (**D**) LEfSe analysis of cecal microbiota (LDA score is greater than 4). Data are presented as mean ± SD, * *p* < 0.05, ** *p* < 0.01.

**Figure 6 microorganisms-10-01281-f006:**
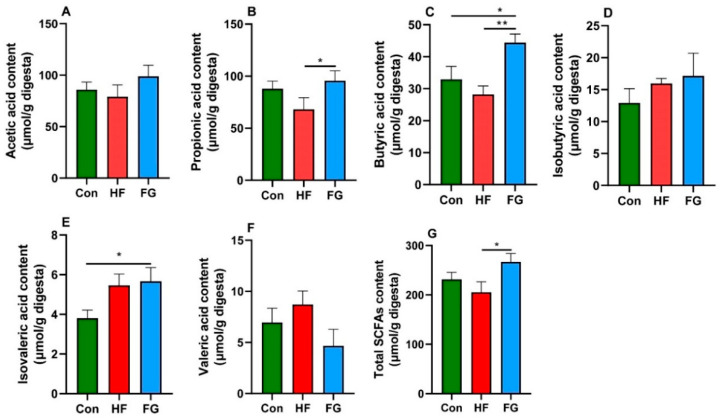
Effects of dietary fat level and GABA supplementation on the cecal concentrations of acetic acid (**A**), propionic acid (**B**), butyric acid (**C**), isobutyric acid (**D**), isovaleric acid (**E**), valeric acid (**F**), and total SCFAs (**G**). Data are presented as mean ± SEM, * *p* < 0.05, ** *p* < 0.01. SCFAs, short-chain fatty acids.

**Figure 7 microorganisms-10-01281-f007:**
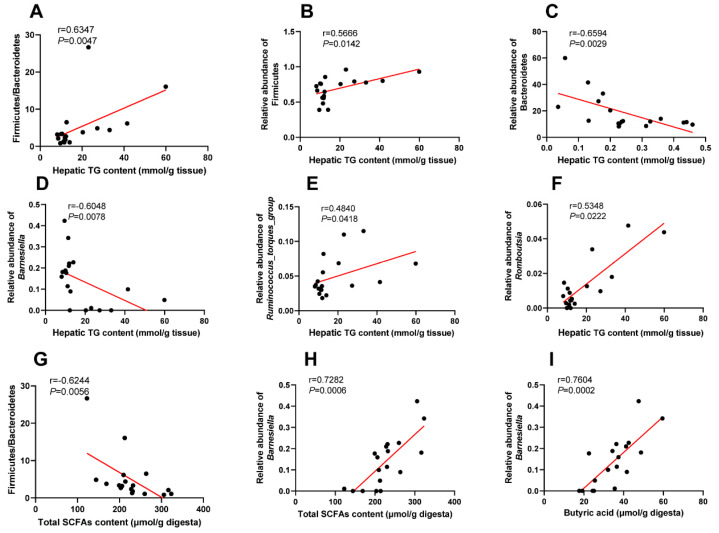
(**A**) the correlation between hepatic TG content and the ratio of Firmicutes/Bacteriodetes; (**B**) the correlation between hepatic TG content and the relative abundance of Firmicutes; (**C**) the correlation between hepatic TG content and the relative abundance of Bacteriodetes; (**D**) the correlation between hepatic TG content and the relative abundance of *Barnesiella*; (**E**) the correlation between hepatic TG content and the relative abundance of *Ruminococcus_torques_group*; (**F**) the correlation between hepatic TG content and the relative abundance of *Romboutsia*; (**G**) the correlation between cecal SCFAs and the ratio of Firmicutes/Bacteriodetes; (**H**) the correlation between cecal SCFAs and the relative abundance of *Barnesiella*; (**I**) the correlation between cecal butyric acid and the relative abundance of *Barnesiella*.

**Table 1 microorganisms-10-01281-t001:** The nutrient component of the diet.

	1–21 d		22–40 d	
Ingredients (g/kg)	Con	HF	Con	HF ^1^
Corn	559.6	514.6	613.7	568.7
Soybean meal	374	374	324	324
Soybean oil	25	70	25	70
Dicalcium phosphate	13	13	12.4	12.4
Calcium carbonate	11.5	11.5	11.2	11.2
Sodium chloride	2.3	2.3	2.2	2.2
Sodium bicarbonate	1.3	1.3	1.5	1.5
DL-Methionine	3.9	3.9	2.5	2.5
L-Lysine	2.0	2.0	/	/
L-Threonine	0.8	0.8	1.1	1.1
Starter premix ^2^	5.0	5.0	/	/
Finisher premix ^3^	/	/	5.0	5.0
Choline chloride	0.4	0.4	0.2	0.2
Nutritional composition (g/kg)				
DM	909.7	921.2	889.8	899.6
CP	227.2	227.8	194.8	189.6
Total fat	55.9	85.3	58.1	94.5
Ash	64.1	65.4	55.8	54.3
ME (MJ/kg)	12.36	13.41	12.58	13.63

^1^ Con = low-fat diet; HF = high-fat diet. ^2^ Starter premix kg^−1^: 3,000,000 IU of vitamin A; 100,000 IU of vitamin D3; 900 mg of vitamin E; 100 mg of vitamin K3; 600 mg of vitamin B1; 330 mg of vitamin B2; 160 mg of vitamin B6; 20 mg of vitamin B12; 20 mg of biotin; 1000 mg of Ca pantothenate acid; 35 mg of folic acid; 15,000 mg of vitamin C; 1000 mg of vitamin B3; 3500 mg of Zn, 9000 mg of Fe; 3000 mg of Mn; 700 mg of Cu; 100 mg of I; 70 mg of Co; 12 mg of Se. ^3^ Finisher premix kg^−1^: 2,500,000 IU of vitamin A; 100,000 IU of vitamin D3; 800 mg of vitamin E; 80 mg of vitamin K3; 400 mg of vitamin B1; 290 mg of vitamin B2; 100 mg of vitamin B6; 20 mg of vitamin B12; 20 mg of biotin; 900 mg of Ca pantothenate acid; 30 mg of folic acid; 15,000 mg of vitamin C; 1000 mg of vitamin B3; 3000 mg of Zn, 8000 mg of Fe; 3000 mg of Mn; 600 mg of Cu; 100 mg of I; 50 mg of Co; 12 mg of Se.

**Table 2 microorganisms-10-01281-t002:** The primer sequences for real-time qPCR.

Target Genes ^1^	Primer Sequences	Source
*PPIA*	F: GTGACTTTACGCGCCACAAC	NM_001166326
	R: TTGCTCGTCTTGCCGTCTTT	
*PPARα*	F: AGGAAATCTACAGGGACA	NM_001001464
	R: GAACCGAGTGAACAGC	
*CPT1α*	F: GGCATTGACCGCCATCTGT	NM_001012898
	R: GAAACACCGTAACCATCATCAGC	
*ACC*	F: GCCTCCGAGAACCCAA	NM_205505
	R: CTGTTGAGATGTGAGACTGT	
*SREBP1*	F: CTACCGCTCATCCATCAACG	NM_204126.2
	R: CTGCTTCAGCTTCTGGTTGC	
*PLIN1*	F: GGCTATGGAGACGGTGGATG	NM_001127439
	R: CTGGCTTGCTCTCCTCTTCC	
*FATP1*	F: CCTTGTTGACTCCGGGTAT	NM_001398142
	R: TGGGCTCTGGTGTTCTTC	
*ATGL*	F: GCTGATCCAGGCCTGTGTCT	NM_001113291
	R: TGGAGGTATCTGCCCACAGTAGA	
*LPL*	F: GACAGCTTGGCACAGTGCAA	NM_205282
	R: CACCCATGGATCACCACAAA	
*FAS*	F: CGTCATCACCGTCTATC	NM_205155.3
	R: GTAGGCTCCTCCCATC	
*GPX1*	F: GACCAACCCGCAGTACATCA	NM_001277853
	R: GAGGTGCGGGCTTTCCTTTA	
*HO1*	F: GGTCCCGAATGAATGCCCTTG	NM_205344.2
	R: ACCGTTCTCCTGGCTCTTGG	
*SOD*	F: GGTCATCCACTTCCAGCAGCAG	NM_205064.2
	R: AAGCCATGATCTCCATCAGACAAGC	
*NRF2*	F: CTGCTAGTGGATGGCGAGAC	NM_001030756
	R: CTCCGAGTTCTCCCCGAAAG	

^1^ The abbreviations of the gene names are shown as follows: *PPIA*: Peptidylprolyl isomerase A; *PPARα*, peroxisome proliferator-activated receptor α; *CPT1α*, carnitine palmitoyl transferase 1α; *ACC*, acetyl-CoA carboxylase; *SREBP1*, sterol regulatory element-binding protein 1; *PLIN1*, Perilipin 1; *FATP1*, fatty acid transport protein 1; *ATGL*, adipose triglyceride lipase; *LPL*, lipoprotein lipase; *FAS*, fatty acid synthase; *GPX1*, glutathione peroxidase 1; *HO1*, heme oxygenase 1; *SOD*, superoxide dismutase; *NRF2*, nuclear factor E2-related factor 2.

**Table 3 microorganisms-10-01281-t003:** Effect of high-fat diet and GABA supplementation on the serum lipid profile in broilers.

Parameters ^1^	Con	HF	FG
HDL-C (mmol/L)	2.47 ± 0.16	2.59 ± 0.16	2.69 ± 0.12
LDL-C (mmol/L)	0.83 ± 0.06	0.68 ± 0.04	0.68 ± 0.07
TC (mmol/L)	3.23 ± 0.17	3.24 ± 0.19	3.24 ± 0.17
TG (mmol/L)	0.561 ± 0.043 ^a^	0.808 ± 0.119 ^b^	0.544 ± 0.052 ^a^

^1^ low-density lipoprotein cholesterol. ^a,b^ Mean with different superscripts in the same row differ significantly (*p* < 0.05).

## Data Availability

The data presented in this study are available on request from the corresponding author.

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
