# Peer review of "Dietary γ-Aminobutyric Acid Supplementation Inhibits High-Fat Diet-Induced Hepatic Steatosis via Modulating Gut Microbiota in Broilers"

_microorganisms, 2022, doi:10.3390/microorganisms10071281_

Round 1
Reviewer 1 Report
The researchers found that GABA supplementation reduced the negative effects of feeding a high fat diet to broilers. This included reducing TG deposition, steatosis, and aided in maintaining a more "healthy" microbiota in the gut.
The results, statistical analysis and presentation in the manuscript are represented well in the figures and tables provided. The data clearly supports the conclusions drawn by the authors. Proper ACUC guidelines were followed.
The results are interesting and would be of interest to readers due to the importance of chronic inflammatory processes associated with feeding high fat diets to poultry. The possible use of GABA as a prebiotic may aid in alleviating such effects.
No further comments.
Author Response
Thank you for your recognition of our research.
Reviewer 2 Report
In my opinion, manuscript entitled ,,Dietary γ-Aminobutyric Acid Supplementation Inhibits High-2 Fat Diet-Induced Hepatic Steatosis via Modulating Gut Micro-3 biota in Broilers” is wel written and can be published in Microorganisms journal. I have only a few suggestions listed below.
- Line 41: GABA (γ-aminobutyric acid)
- Line 159: Tukey-Kramer
M&M: please define an abbreviation for feed with GABA added (probably FG?).
Author Response
Line 41: We have added the full name of GABA.
Line 159: We have revised it to the Tukey-Kramer.
Line 65-66: The feed with GABA added have defined as FG.